# Having to Work from Home: Basic Needs, Well-Being, and Motivation

**DOI:** 10.3390/ijerph18105149

**Published:** 2021-05-13

**Authors:** Hannah M. Schade, Jan Digutsch, Thomas Kleinsorge, Yan Fan

**Affiliations:** Leibniz Research Centre for Working Environment and Human Factors, Technical University Dortmund, Ardeystr. 67, 44139 Dortmund, Germany; digutsch@ifado.de (J.D.); kleinsorge@ifado.de (T.K.); fan@ifado.de (Y.F.)

**Keywords:** work from home, work-related basic needs, work engagement, well-being at work, adjustment to COVID-19 measures, work in the pandemic

## Abstract

During the COVID-19 pandemic, many employees were asked to start working from home for an extended time. The current study investigated how well employees worked and felt in this novel situation by following *n* = 199 German employees—56% of them female, 24% with childcare duties—over the course of two working weeks in which they reported once daily on their well-being (PANAS-20, detachment) and motivation (work engagement, flow). Participants reported on organizational and personal resources (emotional exhaustion, emotion regulation, segmentation preference, role clarity, job control, social support). Importantly, they indicated how well their work-related basic needs, i.e., autonomy, competence, and relatedness, were met when working from home and how these needs had been met in the office. Multilevel models of growth showed that work engagement, flow, affect and detachment were on average positive and improving over the two weeks in study. Higher competence need satisfaction predicted better daily work engagement, flow, and affect. In a network model, we explored associations and dynamics between daily variables. Overall, the results suggest that people adapted well to the novel situation, with their motivation and well-being indicators showing adequate levels and increasing trajectories. Avenues for improving work from home are job control and social support.

## 1. Introduction

In the first months of 2020, the coronavirus pandemic resulted in an unforeseen disruption of many people’s daily life. In order to contain the pandemic, public life came to a halt in many countries around the globe. People were asked to stay at home, schools were closed, public events were banned, ‘non-essential’ production stopped, social distancing was strongly encouraged, and even lockdowns were imposed. In Europe, these measures were put into place in March; governments differed in the exact measures they ordered. For example, France, Italy, and Spain announced strict nationwide lockdowns allowing people only to leave their homes for essential tasks, when carrying a form stating their purpose and time of leaving the house, with lockdown violations being fined drastically [1]. The present study took place in Germany, which also closed schools, daycare facilities, restaurants, gyms, etc., cancelled public events, and imposed travel restrictions. However, Germany did not issue a hard lockdown with a strict confinement order or curfew in the spring of 2020, but instead relied on people’s compliance with social distancing rules—no more than two people (from different households) were allowed to meet at the same time, and violations were fined. Generally, people were strongly encouraged to stay at home and to limit their social contact but were not forbidden to leave the house to take a walk or see a friend [1]. Thus, the intervention in the private life was not as severe as in other countries, but nevertheless daily routines were disrupted to a large extent. Many employees were ordered short-time work or to work from home, with about one-quarter of the German workforce working from home in April of 2020 [2]. Importantly, many people had to simultaneously take care of their underage children who normally would have been in daycare or school, further complicating the challenge of suddenly having to work from home, which happened for many without any transition period.

Thus, the coronavirus pandemic thus resulted in an abrupt change in daily routines and working conditions for many employees in Germany. From a psychological point of view, this constituted a quasi-experimental mass intervention that challenged people’s self-regulation capacities. Because of the cross-the-board nature of this challenge, it hit people with different personal preconditions and social resources to cope with this challenge in a more or less successful manner. We considered this setting as an opportunity to assess how employees would fare in this novel situation, to follow their adaptation to this challenge, and to investigate the conditions that would help or hinder successful adaptation. As work psychologists, we focused specifically on the difference between working in the office and working from home.

We reasoned that adaptation to this novel situation may partially depend on the degree to which work from home allows satisfaction of the basic needs, i.e., the need for competence, autonomy, and relatedness [3,4], which humans need in order to thrive and be motivated according to Self-Determination Theory [3]. That is, people need to experience themselves as self-determined active agents (autonomy) that are able to produce desired outcomes (competence) and connect meaningfully with others (relatedness). These needs are described as ‘basic’ in the sense of applying to any person or context [3]. In a context that fulfills these needs, humans develop intrinsic motivation, they feel good and work well. Thus, it makes sense that the concept of basic need satisfaction has also been applied to the work sphere [4]. Specifically, in order to feel motivated at work, employees need to experience the ability to (a) choose and decide in which way to tackle work tasks (autonomy), (b) master work tasks well (competence), and (c) have meaningful interactions with colleagues (relatedness). It deemed us worthwhile to ask employees in what way work-related need satisfaction may have changed by working from home and how work-from-home autonomy, competence, and relatedness experience may have affected everyday motivation and well-being.

We started this broad, exploratory study with the overarching goal of understanding how well people felt and worked when suddenly having to work from home. We used the lens of Self-Determination Theory [3] to understand the interplay of needs, resources, and motivation in everyday life in the home office. To shed light on this obviously multi-faceted phenomenon, we approached work-related well-being from several angles:

First, we assessed how the work-related needs of autonomy, competence, and relatedness [4] were met at the time of the study, working from home, as compared to (retrospective ratings of) how autonomous, competent, and related to colleagues participants had felt when they had been going to the office regularly. Specifically, we assumed that autonomy may be boosted when working from home, competence might remain unaffected on average, and relatedness should be decreased. While we had no solid ground for theory-driven specific hypotheses, we reasoned that while working from home goes along with a loss of social proximity to colleagues that thwarts work-related relatedness for most people, working from home may incur greater autonomy in the work domain, at least for some people. In part, these differences may be attributable to organizational or personal resources that may help or hinder the experience of autonomy, competence, and relatedness when working from home, e.g., whether competence can be experienced working from home may depend on the level of role clarity, i.e., knowing what to do without the need to ask superiors for guidance regularly [5]. This is exemplar for our general interest in exploring the role of individual and organizational resources that might help or hinder need satisfaction when working from home.

Second, we wanted to see whether and how the (lack of) fulfilment of these needs would impact upon the ease of transitioning into working from home. To this end, we gathered daily self-reports of affective well-being, motivation (work engagement, flow experience), and detachment (from work in the free time) over the course of two work weeks. Greater fulfilment of the basic work-related needs has been shown to predict job satisfaction, intrinsic motivation, and performance [4]. We thus expected greater need fulfilment in the home office to yield better work engagement and flow as markers of motivation, as well as better emotional experience and greater psychological detachment as markers of well-being; these questions were explored in multilevel models of change. As an example of the resources people brought into the novel situation, we speculated that people who care more to separate work and life would be better able to detach from work in the evenings. Beyond the effect of needs and resources, the two-week daily reports were used to investigate general dynamics of adaptation to the home office. First, we wanted to know how well employees felt and worked, and in particular, how well they were able to detach from work when it cannot be ‘left at the office’. Second, we investigated whether that got better or worse over time in the home office, as we expected people to increasingly find ways of constructively adapting to the novel situation.

Third, we wanted shed light on the everyday life dynamics of working and thriving when working from home. This was explored by means of dynamic network modeling [6,7] describing lead-lag (from one day to the next) and contemporaneous (same-day) as well as between-person (mean-level) associations between work engagement, flow, detachment, and positive and negative affect. To gather the data necessary to investigate these questions, we approached people that had transitioned into the home office and asked them to partake in our study on well-being and motivation when working from home. Because we were mainly interested in the adaptation to an unfamiliar situation, we restricted our sample to employees who were not used to work from home to an extensive degree, despite knowing that, of course, people would have been working from home at least some days before starting participating in the study. After assessing a number of background variables, we followed participants’ trajectory of adaptation in terms of productivity and well-being across two working weeks by means of quantitative daily online diaries. 

## 2. Materials and Methods

### 2.1. Participants and Procedure

The present study was conducted online during April and May of 2020, the peak of the first wave of the COVID-19 pandemic in Germany. In order to limit the spread of the virus, the government asked many people to work from home. We recruited these people for participation in our study via our institutional website, Twitter, and mail distribution lists to employees of the Dortmund city council and a regional insurance company. Criteria for election were that participants currently worked from home for most days of the week but had previously worked from home on no more than one day per week and were able to do more or less the same tasks from home as they had done in the office. For completion of more than 70% of the questionnaires, participants were compensated with EUR 37.50. The study was approved by our institute’s ethics committee (permit no. 160). Once participants had signed up for the study and given consent to its modalities, they were sent the link to a background questionnaire assessing their demographics and work environment as well as organizational and personal resources. In addition, we asked them to indicate twice how well their basic work-related needs for autonomy, competence, and relatedness were satisfied: retrospectively for when they had been working in the office, and how they were satisfied in the current situation working from home. We are aware that retrospective assessment is not perfect, with memory being subject to recall bias [8], although studies suggest that recall bias is rather small for estimating well-being in the recent past [9]. The daily questionnaires assessing their daily well-being and motivation started on the subsequent Monday and lasted for two work weeks—most daily diary studies span one to three weeks in the hope to capture enough fluctuation but not to risk too much drop-out when participants grow tired of the study. Questionnaires were sent out once on each workday at the time participants had indicated they would typically finish their working day, resulting in 1652 observations of the daily variables. 

Participants indicated *to what percentage they had worked from home before the pandemic* (*M* = 20.35, *SD* = 30.86). In order to allow for this study to tap into adaptation dynamics to a somewhat novel situation, we reduced the total sample of *n* = 328 to people that worked no more than 30% of their work hours from home before the coronavirus situation, which led to the exclusion of 59 persons. Another 57 persons were excluded as they reported working less than 30 h per week, and our interest was in the well-being and needs of people whose work week was primarily determined by work conditions. Lastly, 13 people were excluded due to missing data on the basic needs questionnaire, resulting in a total sample of 199 people who had been completing on average 6.86% (*SD* = 8.83) of their weekly work from home before the pandemic. It is this sample of 199 participants that is described in the following: 

A total of 57% of our sample was *female*, i.e., it consisted of 114 women and 85 men with a mean *age* of 39.5 (*SD* = 11.4, range: 21–65). The mean *household size* of our participants was 2.4 (*SD* = 1.19). Our sample was working on average 40.6 *work hours per week* (*SD* = 5.03, range: 32–58), which was on average 1.59 h (*SD*= 4.97) more than agreed per work contract. Most of our sample was able to choose freely when to start/end the workday (90% yes); moreover, 50% of the sample indicated that they occasionally use a smartphone to do work-related things in the evening. Most of the sample reported working on the PC or communication with customers as their primary tasks, many of them in the public sector, in IT/communication, or science. On average, our sample had spent approximately 10 *years with the current employer* (*M* = 9.4, *SD* = 9.59) and 20 *days working from home at the moment of starting participation in the study* (*M* = 20.5, *SD* = 9.36). Participants found their workspace at home a rather *appropriate place to work* (*M* = 3.83, *SD* = 1.09, on the usual 5-point scale). Given that schools and pre-school facilities were closed, we asked participants to indicate to what extent (%) they were currently responsible for taking care of a child or children in the household (*M* = 11.8, *SD* = 25.0); 152 people, 76% of the sample included in the analyses reported 0 *childcare duties*. Interestingly, about the same number of people of both genders reported having childcare duties (24 women, 23 men).

### 2.2. Measures

The full list of questionnaires, as well as the data and pre-registered hypotheses, can be found on osf.io (https://osf.io/r2j4c/, accessed on 11 May 2021). The measures reported here are those used for the analyses for this paper (mainly based on pre-registered hypotheses). Unless otherwise specified, answers were given on a scale from 1–5, with higher scores indicating a higher level of the assessed construct. For demographics, please consult the last paragraph of the previous section describing participants and procedure.

Three types of data were assessed: first, work-related basic need [4] satisfaction (Section 2.2.1) was expected to have changed by being sent to work from home, and thus people were asked to rate their current need satisfaction working from home, as well as to remember their need satisfaction when they had been working in the office; second, background questionnaires (Section 2.2.2) describing (largely) stable personal and organizational resources were assessed only once at the beginning of the study; and third, we assessed everyday well-being and motivation when working from home, asking the same questions 10 times, i.e., at the end of each work day, over the course of two weeks—this repeated assessment allowed us to understand fluctuation in and development of well-being and motivation, as well as to explore their associations in a network model.

#### 2.2.1. Work-Related Basic Needs Satisfaction

Participants filled out the *Work-Related Basic Needs* scale [4] twice: First, they were asked to think back how the three needs had been met when they still had been going to the office before the pandemic hit and to give a retrospective estimation of that time, and then we asked them to picture their acute situation and rate how the three needs were met concurrently, i.e., at the time of working from home. Values are presented in Table 1. For comparison, in all samples, our lab collected this information in the years before the pandemic: autonomy levels were at *M* = 3.35 (*SD* = 1.2), competence levels at *M* = 4.09 (*SD* = 0.93), and relatedness levels at *M* = 3.89 (*SD* = 1.23).

#### 2.2.2. Background Questionnaires: Personal and Organizational Resources

Participants reported on an array of resources that may help or hinder coping with the novel situation and having a positive work experience in general: role clarity [5], job control [10], social support by colleagues [11], emotional exhaustion by the Maslach Burnout inventory [12], segmentation preference [13], and the tendency to reappraise [14,15], as illustrated in Table 2. These measures were assessed once only upon entry in the study, as they are considered relatively stable. We gathered this information in order to explain variance in need satisfaction when working from home, as well as variance in daily work-from-home levels of motivation and well-being.

#### 2.2.3. Daily Measures

The daily measures were assessed once a day at the time people reported typically ending their workday. For means and sample items, see Table 3. *Affect* was assessed using the PANAS-20 [16,17], which include the affect items (German in brackets) active (“aktiv”), interested (“interessiert”), excited (“freudig erregt”), strong (“stark”), inspired (“angeregt”), proud (“stolz”), enthusiastic (“begeistert”), alert (“wach”), determined (“entschlossen”), and attentive (“aufmerksam”) for positive affect, and upset (“verärgert”), distressed (“bekümmert”), hostile (“feindselig”), irritable (“gereizt“), nervous (“nervös“), confused/jittery (“durcheinander“), afraid (“ängstlich“), guilty (“schuldig“), scared (“erschrocken”), and ashamed (“beschämt“) for negative affect. Items were averaged into positive affect and negative affect composites for analyses. *Psychological detachment* was assessed with the detachment subscale of the Recovery Experience Questionnaire [18], *Flow* was measured by the 4 items capturing the absorption factor from the flow short scale [19], and *Work Engagement* was assessed by the Utrecht Work Engagement questionnaire [20] as adapted to daily assessment.

### 2.3. Statistical Analyses

#### 2.3.1. Need Satisfaction

Need satisfaction when (a) working from home (at the moment) vs. (b) when still working in the office (before the pandemic) was compared with paired *t*-tests for dependent samples (see Figure 1). The role of background variables for need satisfaction when working from home was explored in regressing them onto work from home-levels of each need’s satisfaction (see Table 4).

#### 2.3.2. Adaptation Dynamics: Multilevel Model of Change

In order to assess the degree of adaptation to the new situation, we applied multilevel growth curve models to the variables assessed once-daily over the course of two work weeks. The models were specified at the within-person level as, for the example of detachment,
(1)$$Detachment_{ti}=\beta_{0i}+\beta_{1i}\left(time_{ti}\right)+e_{ti}$$
where $Detachment_{ti}$ represents individual i’s detachment at time t; $\beta_{0i}$ is a person-specific intercept parameter; $\beta_{1i}$ is a person-specific linear slope parameter that characterizes the rate of change per day in study; and *e_ti_* is residual error. Person-specific intercept ($\beta_{0i}$) and linear ($\beta_{1i}$) parameter were modelled at the between-person level including the person-specific levels of need fulfilment when working from home that were centered at the grand mean, specifically, as
(2)$$\beta_{0i}=\gamma_{00}+\gamma_{01}\left(autonomy_{i}\right)+\gamma_{02}\left(competence\right)+\gamma_{03}\left(relatedness\right)+u_{0i}$$
(3)$$\beta_{1i}=\gamma_{10}+\gamma_{11}\left(time\right)+u_{1i}$$

The covariates were grand mean-centered so that $\gamma_{00}$ and $\gamma_{10}$ indicated the average trajectory across all individuals. Residual between-persons differences *u*_0*i*_ and *u*_1*i*_ were assumed to be multivariate normally distributed, correlated with each other, and uncorrelated with the residual errors, *eti*. Models were estimated with the *lme4* package in R [21], with incomplete data treated as missing at random. As the daily variables—detachment, work engagement, flow, and positive and negative affect—could not clearly be divided into predictors and outcomes on a theoretical basis, we decided to use all of them next to each other as dependent variables in separate multilevel models that describe the experience of working from home as a function of time and background variables. To investigate the relationships between the daily variables, we opted to use dynamic network modelling, which does not require a priori assumptions about the direction of associations between variables, but instead simultaneously estimates partial regressions for any pair of variables. 

#### 2.3.3. Dynamic Network Analysis

To explore the associations among the daily variables, we performed dynamic network analysis using the *panel-lvgvar* function in the psychonetric package in R [6,7]. Dynamic network models are a type of multilevel graphical vector autoregression (multilevel GVAR) model between latent variables that build upon Gaussian graphical models but account for the clustering of repeated measures data within individuals [7]. Gaussian graphical models [22] are great for exploring associations between a set of variables, as they are undirected network models presenting partial correlation coefficients between all variables in the model. Network analysis features each variable as a “node” in the network diagram; the association among the nodes are indicated by “edges”, which represent the association of a pair of variables while controlling for all other variables. The Graphical Vector-Autoregression Model (GVAR) extends the GGM and accounts for the temporal dependency of the variables, which can be described as the following regression formula:$$y_{t}=\mu +B\left(y_{t-1}-\mu \right)+\zeta_{t},\zeta_{t}\sim N\left(0,\Sigma^{\left(\zeta \right)}\right)$$
where $y_{t}$ represents a fixed response pattern at timepoint t for a single subject (ignoring the subscript p for subject), and μ  represents the mean vector. The matrix B represents the temporal within-subject network and is standardized to partial directed correlation. The temporal network shows how each variable at the current timepoint predicted all other variables at the next timepoint, i.e., each variable at timepoint t was regressed on all the variables at timepoint t − 1. 

The matrix $\Sigma^{\left(\zeta \right)}$ is a GGM representing the within-subject contemporaneous network, which estimates the association between variables at time point t while accounting for the temporal effects at t − 1.

Finally, the within-subject mean values are calculated to estimate another GGM—the between-subject network—representing the covariance between the average level of the variables of different subjects.

In the current model, we used daily measures of work engagement, flow, detachment, positive affect, and negative affect. Drawing from the same underlying statistical model, we estimated three types of networks: temporal, contemporaneous, and between-persons. We estimated the stability of the network model by performing 1000 times 25% case-drop bootstraps [23]. The level of stability is indicated by the number of times that each edge is included in the 1000 bootstrap samples (see Table A1 and Table A2).

## 3. Results

### 3.1. Autonomy, Competence, and Relatedness

The satisfaction of the three work-related basic needs when working in the office vs. when working from home are illustrated in Figure 1. As expected, relatedness to co-workers was perceived as significantly lower when working from home than in the ratings of how it had been when working in the office, *t*(198) = 17.1, *p* < 0.001. There was no statistically significant difference between work environments for autonomy, *t*(198) = −0.89, *p* = 0.38, or competence, *t*(198) = 1.38, *p* = 0.17.

### 3.2. Background Variables

Background variables were related to basic needs as reported in Table 4, which presents the results from three partial regression models predicting work-from-home levels of autonomy, competence, and relatedness. Autonomy and competence when working from home were both positively predicted by how autonomous or competent, respectively, the employee had felt in the office before, while relatedness to colleagues during office times had no significant influence on how connected participants felt to their colleagues when working from home. Job control improved autonomy and competence fulfilment when working from home as expected. A tendency to reappraise problematic situations helped all three needs’ satisfaction in the home office, as expected. Relatedness levels were positively associated with social support and age. Unexpectedly, competence working from home was slightly negatively associated with office time autonomy levels. With age, competence and relatedness needs were met better. Role clarity, emotional exhaustion, segmentation preference, and childcare duties did not influence any need’s fulfilment working from home in our sample.

### 3.3. Daily Measures and Their Development over Time

On average, all daily indicators of well-being and motivation were at acceptable or good levels and, to a small extent, improved over the two weeks in study, speaking for participants being able to adapt to the novel situation more and more and mostly do well when working from home. Table 5 lists the estimates of work engagement, detachment, flow, and positive and negative affect, including the effect need satisfaction has on their levels. Competence experience when working from home emerged as significant predictor of work engagement and flow, as well as positive and negative affect. 

Estimates reveal substantial interindividual variance remaining; background variables did not explain considerable variance and are thus not included in the models reported in the table. Two observations are of note: Detachment was indeed significantly (both *p*’s < 0.01) improved by greater levels of segmentation preference (*β* = 0.24) and social support (*β* = 0.12) when adding these (centered) predictors to the model reported in the equation and table. Including childcare duties in the models presented here did not yield any significant effect, with or without having it interact with gender.

### 3.4. Network Models

We first estimated a model including all edges in the temporal, contemporaneous, and between-subject network using the *panel-lvgvar* function in the psychometric package in R (alpha = 0.01, uncorrected). We then performed model pruning using the *modelsearch* function in the psychonetric package, i.e., we cleaned and reduced the network to non-redundant, critical values. The pruned model showed good model fit, with an RMSEA of 0.094 (95% CI 0.090–0.098), and good incremental fit (NFI = 0.69, PNFI = 0.73, TLI = 0.79, NNFI = 0.79, RFI = 0.71, IFI = 0.78, RNI = 0.78, CFI = 0.78). Figure 2 illustrates network structures in the pruned model of daily variables included in the network model, i.e., work engagement, flow, detachment, and positive (PA) and negative affect (NA). The appendix includes Table A1, which shows the numeric estimates of the standardized network parameters. 

To examine the stability of the network structure of the daily variables, we drew 1000 case-drop bootstrap samples from the original sample. In each bootstrap sample, 25% of the cases were dropped at random. We estimated the temporal, contemporaneous, and between-subject networks again with the bootstrap samples and counted the number of times each edge was included in the 1000 estimations (see Figure 3 and Table A2 in the Appendix A). We found high stability in the contemporaneous and the between-subject networks, and moderate stability in the temporal network. Edges that were included in the original pruned model were also likely to be included in the bootstrapped analyses. In the following, we only report on the edges that were found in at least 50% of the bootstrap samples, in order to ensure the results discussed are likely more than flukes.

#### 3.4.1. The Temporal Network

The temporal network indicates how one variable on day *n* predicts the other variable on day *n* + 1 while controlling for the temporal relationships with all other variables in the model, as well as controlling for the contemporaneous associations of the variables, which are estimated concurrently and presented in the contemporaneous model described below. In our model, flow very robustly predicted work engagement (*r* = 0.13) on the next day. Negative affect slightly decreased next-day levels of work engagement (*r* = −0.06) and detachment (*r* = −0.08). Work engagement was the only construct without significant autocorrelation; all others had significant carry-over effects from day to day (detachment: AR = 0.22, negative affect: AR = 0.19; flow: AR = 0.14, positive affect: AR = 0.1).

#### 3.4.2. The Contemporaneous Model

The contemporaneous model estimates partial correlations between the constructs at the same point in time, controlling for the temporal associations that are estimated concurrently (see above). Here, flow and work engagement showed a strong correlation again (*r* = 0.67), and work engagement and positive affect (PA) were also considerably associated (*r* = 0.39). Negative affect (NA) was again negatively associated with detachment (−0.25), PA (−0.24), and work engagement (−0.14). Small but robust associations were found for work engagement and detachment (0.12), as well as for flow with NA (0.12) and PA (0.12).

#### 3.4.3. The Between-Subject Model

At the between-subject level, work engagement was again strongly positively associated with flow (*r* = 0.74) and PA (*r* = 0.55). People who generally reported higher detachment also generally reported lower NA (*r* = −0.39) and higher PA (*r* = 0.19).

## 4. Discussion

This study set out to shed light on work from home during the pandemic, with a focus on work-related basic need satisfaction as well as adaptation dynamics. Results suggest that working from home primarily decreases relatedness to coworkers, while autonomy and competence stay at levels comparable to before on average. Competence experience, however, is the one need predicting better daily work engagement, flow, and affect working from home. All these daily variables improved over the two weeks in study, corroborating our hypothesis that employees adapt well to working from home and increasingly find their ways of making it work. This also holds for detachment, which is of special interest as it is often discussed to be jeopardized most by working from home. We thus conclude that participants were able to function and feel pretty well when working from home. A temporary network model showed the dynamics between detachment, work engagement, flow, and affect, providing proof of concept for the use of dynamic network to describe everyday dynamics between repeatedly assessed constructs, which deems us a fruitful route for future research.

### 4.1. Work-Related Basic Needs

It is not surprising that relatedness to colleagues is the work-related need most affected by working from home. The tasks to be completed stay the same or comparable for those who can do their job from home, the main difference is the physical and social work environment. This change is so drastic that there is literally no correlation with previous office-time levels of relatedness to colleagues, which is the exception for repeatedly assessed constructs and not true for autonomy and competence, which show at least moderate auto-correlation. While this alerts us to the problem that positive interactions with colleagues cannot be simply be transferred from the office to a virtual team setting and specific preparation or intervention might be required, luckily, we do not have to conclude that none of the social resources from office times persist into a work-from-home setting: social support by colleagues contributed to relatedness need satisfaction working from home, i.e., more supportive colleagues made participants feel less alone in the home office. This should be taken seriously for any virtual team or longer-term work-from-home arrangement; employers might want to create an infrastructure and climate that encourages supportive interactions between employees, particularly given that greater social support predicted better daily detachment from work. That is, knowing my colleagues will have my back if necessary makes it easier to forget work worries in the evening. Note that in our data, relatedness (to colleagues) did not translate into daily detachment, motivation, or affect, not even when relatedness was the only predictor included in the model. This might have partially been due to family and roommates being able to compensate for the absence of coworkers. However, it might also underline the importance of the quality of social relationships: strong supportive relations to colleagues may be a more important buffer against work-related stress than friendly interactions and regular small-talk, as much as people report missing this when not being allowed to go to the office.

Competence needs were expected to be met in both office and home equally, which was the case for the average person in our sample. As expected, greater levels of job control translated into greater levels of competence, and competence experience when working from home did improve flow, work engagement, and positive affect, and reduced negative affect to a considerable extent. This centrality of competence need satisfaction for daily well-being and motivation was unexpected but could be made sense of by considering competence needs as *the* need people expect to satisfy by work means, while the satisfaction of relatedness and autonomy may also be achieved in other life domains. It may pay off for employers and leaders to do everything in their power to create the prerequisites so that their staff can experience themselves as competent in their work, also when it is completed from home. One step in that direction may be granting employees higher levels of job control, i.e., to allow and empower them to decide for themselves how and when to complete a task, instead of strict protocols or micro-management, which may be particularly frustrating and limiting in remote work settings. This seems to be difficult for some employers and team leaders, who have trouble letting go of control in the work from home setting, while this exact thing might unleash the desired productivity. In general, however, the level of job control reported by our sample seems rather satisfactory (3.33 on a 5-point scale) and is comparable to pre-coronavirus samples acquired by our institute with an average mean of 3.18. This hesitation to increase the level of job control when employees work from home is a recurring topic in the media these days and might also explain why working from home did not increase autonomy need satisfaction, as we had originally expected. Of course, autonomy experience might be greater in other work-from-home settings than this one, given that in this case, working from home was not an autonomous decision in and of itself, but was forced onto workers due to the pandemic. Moreover, there was very little time to prepare the transition, and therefore in many cases the necessary hard- and software to enable appropriate work from home still had to be bought and set up. Dealing with these things was not optional nor could it be postponed; thus, having to address the transition from office to home office might have yielded lower autonomy levels in our sample than work from home might typically show when all necessary hardware, software, and routines are up and running, and employees can freely choose whether to go to the office on that day or to work from home.

The (small) negative relation of previous autonomy levels and concurrent competence need satisfaction was startling. It might hint at people with more autonomy having less strict scripts of how to go about completing their job, and thus having to master even more self-organization than already required by merely working from home, which might be overwhelming in an early stage of the transition. The negative relation of previous autonomy and current competence might partly be due to people in leading positions (higher autonomy) who struggle at first with finding good ways of leading their team remotely (lower competence experience). It would be interesting to see whether this association indeed vanishes with more experience of working from home. 

### 4.2. Background Variables: Avenues for Interventions

As per individual predictors, it is not surprising that older age goes along with greater competence need satisfaction, as most often it comes with greater experience in the field. It is reassuring that working from home, which most often implies the use of modern technology, did not undo the competence advantage of older, more experienced employees. Older employees had the additional advantage of reporting a higher satisfaction of their relatedness needs—this was likely due to older people needing less contact with people outside their inner circle in order to feel satisfied with their social relations, as posited by socio-emotional selectivity theory and often corroborated [24]. It is not surprising either that a tendency to reappraise challenging situations contributes to greater need satisfaction in this work from home setting and pandemic, underlining once again the importance and power of the *subjective* appraisal of any situation. As per individual resources that shape the daily experience, the straightforward expectation that a greater desire and ability to separate work and leisure contributes to better detachment when working from home was corroborated. We do note that the wording of the segmentation preference scale [13] assesses more of an ability than a preference to separate life spheres and should thus be viewed as a resource rather than a necessity. People lacking this resource, i.e., people low in segmentation preference, can be helped, as these have been shown to profit particularly from a mindfulness intervention targeted at increasing detachment [25]. That childcare duties did not affect need satisfaction or daily work and well-being came as a surprise to us, given that other studies have found parents working from home to be under particular pressure during the pandemic, with mothers often shouldering the greater part of the additional care work [26,27,28]. While of course only one-quarter of our sample reported any childcare duties, it might also be the case that parents that found the time to partake in our study were not so consumed by work and childcare duties than other parents might have been, with many factors playing a role, for instance, age and personality of the child or children. If anything, we found tendencies of negative affect, particularly aggression, to be heightened in the parents, particularly mothers, that had childcare duties next to their work from home commitment. For thorough conclusions about the situation for parents, other publications should be consulted [26,27,28]. We do contend though that for conclusions not about life in a pandemic, but more regular work from home settings, lacking childcare facilities should, hopefully, not be an issue.

### 4.3. Network Analyses: Avenues for Future Research

Our network analyses allowed us to peek into the dynamics between the daily indicators of well-being and motivation. In the temporary, contemporaneous, and between-person networks, common themes were the positive association of work engagement and flow as well as the negative interrelation among negative affect and detachment. The temporal network suggests that this association mainly comes about by flow experience on one day improving next-day work engagement, and negative affect on one day making detachment on the next day harder. These results of the contemporaneous network were mirrored in the between-person network, i.e., people with more flow experience on average tended to report greater work engagement on average, and people who tended to experience more negative affect than others tended to have more trouble detaching from their job than others. An unexpected but robust yet small association emerged between negative affect and flow in the contemporaneous model, which was mirrored by a not-so robust (found in 40% out of 1000 times bootstraps) small negative effect of positive affect on flow of the next day. This may be a fluke in our sample but would be in line with the notion that work engagement may not be characterized and supported by feelings of pure positivity, but more by the co-occurrence or shift from negative to positive affect (cf. the affective shift model of work engagement [29]).

The associations explored in the present network models are not particularly novel, nor do they intend to be. Instead, we intended to use them to exemplify how Gaussian graphical models [22] are generally useful for exploring associations, as they are undirected network models presenting partial correlation coefficients between all variables in the model, and dynamic network models are extending these graphical models to repeated measures data [7]. This novel approach offers promising avenues for exploring daily diary data, given that they are estimating the (partial) associations between all variables in the model at the same time without requiring a priori hypotheses. This tool may develop its full potential particularly when used together with more confirmatory approaches such as structural equation modelling. 

### 4.4. Limitations

The present study represents an attempt to capture adaptation to an unfamiliar working situation imposed by regulatory measures to counter the COVID-19 pandemic. Due to the fact that we had to set up the study when this development was already under way, it is essential to first determine inasmuch we were able to track adaptation to working from home as a dynamic process rather than comparing two different working environments in a steady state. Because our participants already worked about 20 days from home on average when entering into the study, it may be conjectured that their adaptation to working from home had already taken place at this point in time. However, the trajectories observed for indicators of well-being and motivation speak against the assumption that participants already reached a steady state. Furthermore, taking into account the number of days already working from home in the growth models did not significantly increase explained variance, it seems safe to conclude that at least part of adaptation to a new situation was still under way when our study stepped in. 

Another goal of the study was to describe the difference in basic work-related need satisfaction between the work environments at home and at the office. However, at the point the study started, people were already working from home, and thus had to rely on their memory to describe how autonomous, competent, and connected with their colleagues they had felt *before* leaving the office. This evidently opens way for any bias affecting the validity of the estimates [8,9], e.g., participants might have idealized the (recent) past of going to the office in the light of the current isolation at home. While this is a concern to be taken seriously into account whenever interpreting the present results, we feel confident that no systematic bias skewed the data across all people. Contributing to this confidence is the wealth of data our lab has accumulated over the past years on the constructs assessed, e.g., we asked participants to rate their autonomy, competence, and relatedness to colleagues long before the pandemic and large-scale experimenting with work-from-home arrangements. While these data were obtained for samples other the one reported here, its means for office-based need satisfaction are in the ballpark of our sample’s retrospective need satisfaction estimation. More specifically, comparing the descriptives of our study with those of previous studies suggest that, in general, retrospective ratings were in the ballpark of in situ data of pre-pandemic need satisfaction in the office. People may have slightly overestimated how connected they had previously felt to their colleagues as they were missing them during pandemic isolation, but work-from-home levels of relatedness to colleagues were rampantly lower not only than retrospective reports but also comparison sample, pre-pandemic levels of relatedness in the office. Also, participants may have overestimated how much autonomy they had when still going to the office, but that could also be due to people with higher levels of autonomy in general being allowed to work from home (bear in mind that only about one-quarter of the German workforce worked from home during the first ‘lockdown’). 

The COVID-19 pandemic affected participants’ lives in more ways than just by working from home. Many everyday routines were disrupted, from the availability of childcare and schooling to being able to go to the gym or visit larger groups of friends. Thus, this study may not be interpreted as a general study of work from home that is necessarily generalizable to other circumstances; work-related relatedness may be lowered less when being able to meet colleagues and friends outside the workhours, and autonomy working from home may be much higher when people are not forced to do so abruptly by a pandemic. However, given that the pandemic did not render the situation dramatic in the time and place the study took place, it still deems us comparable to non-pandemic work from home. Most of the participants were working in fields such as the public sector which did not face a coronavirus-related threat of unemployment; also, there was no strict curfew in Germany, people were still allowed to see members of one other household, and to leave the house without a permit, thus facing lesser restrictions than other regions in Europe and the world. It is interesting that the German government seemed to use the principles of self-determination theory to tackle the spread of the virus, relying more on citizen’s intrinsic motivation to practice social distancing rather than forcing compliance by restraining autonomy.

## 5. Conclusions

All in all, our study contributes to our understanding of work from home and paints a reasonably positive picture for work-from-home well-being and motivation—workers seem well able to adapt to the changed work environment, be productive and thrive in general, and even to detach healthily. Making sure employees can feel competent even when working from home, e.g., by granting them more control over how and when to complete their to tasks, may further boost their motivation and well-being. Only relatedness to co-workers is considerably affected by not going to the office, which need not be a problem—what matters more than regular friendly interactions with colleagues seems to be knowing they can be relied upon in times of need. Thus, creating an atmosphere in which support provision between colleagues is endorsed seems warranted, particularly when considering long-term work from home arrangements. Employers need not fear the lack of control over their employees when these work from home, as long as they instill intrinsic motivation by enabling employees to feel autonomous, competent, and connected while working, wherever the workplace.

## Figures and Tables

**Figure 1 ijerph-18-05149-f001:**
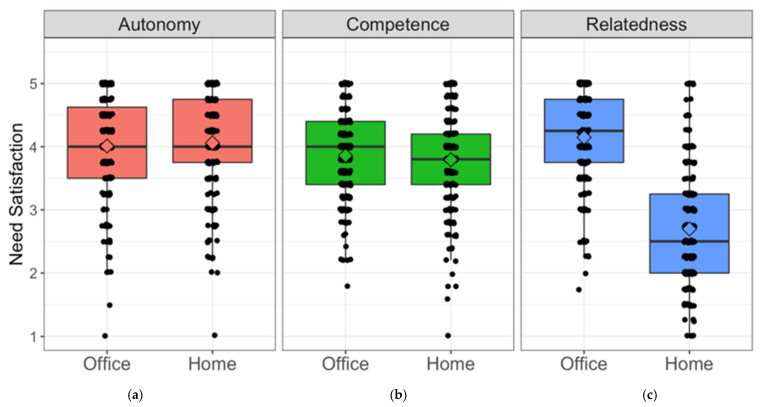
Comparing the basic needs’ fulfilment before the pandemic, i.e., when still working primarily in the office vs. when working from home, showing (**a**) autonomy stays on average the same, (**b**) competence stays on average the same, (**c**) relatedness is substantially lowered when working from home. Boxplots represent the range of the values, with vertical lines indicating the respective standard deviation. Horizontal lines within the boxplots illustrate the arithmetic mean, and diamonds represent the median.

**Figure 2 ijerph-18-05149-f002:**
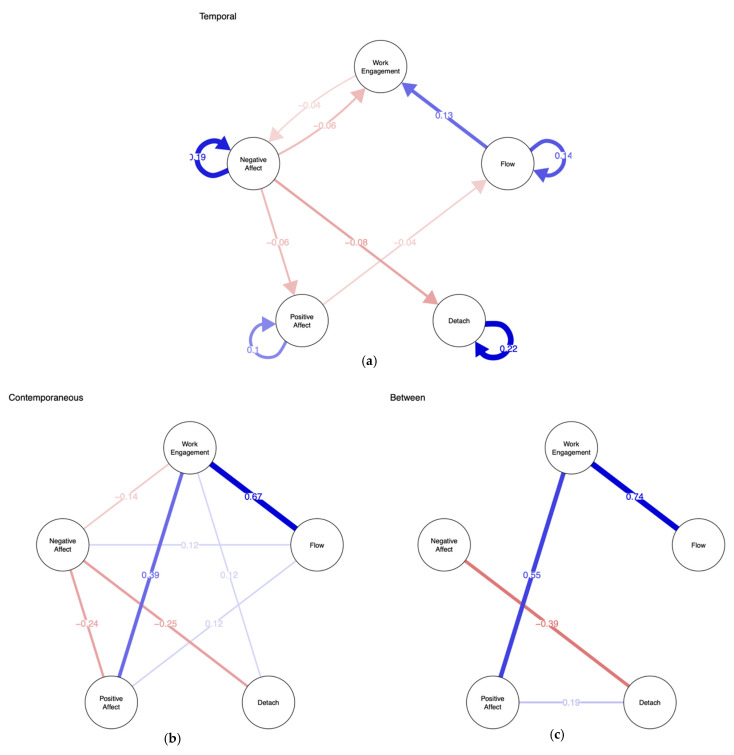
Results of the temporal network analysis of the daily variables work engagement, flow, detachment from work, positive affect and negative affect. Blue lines represent positive associations; red lines negative associations. (**a**) The estimated temporal network, standardized to partial directed correlations, and the edges with arrow indicate how one variable on day *n* predicts the other on day *n* + 1; (**b**) the estimated contemporaneous partial correlation network, and the edges indicate association between two variables on the same day; (**c**) the estimated between-subjects partial correlation network, and the edges indicate cross-subject association among the daily variables.

**Figure 3 ijerph-18-05149-f003:**
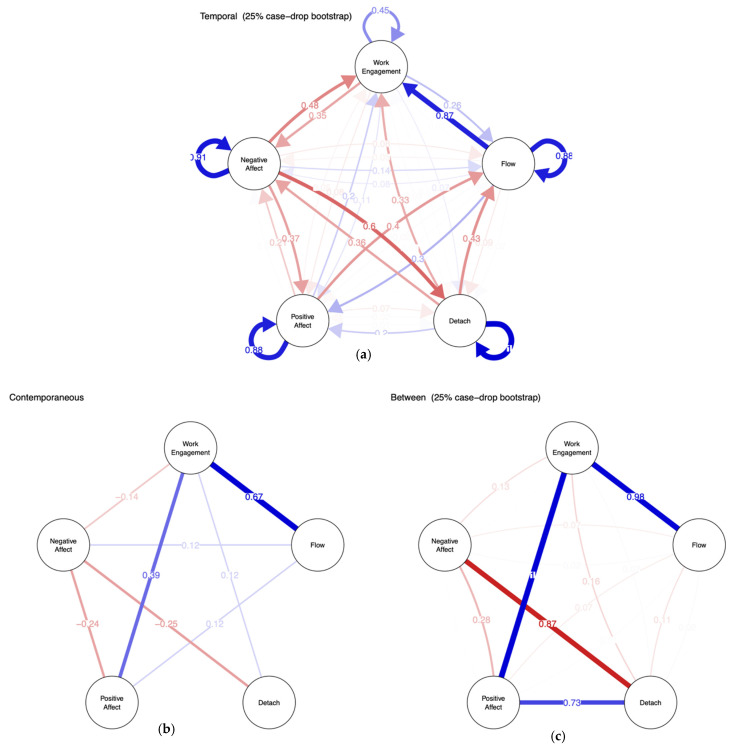
Stability measures of the temporal (**a**), contemporaneous (**b**), and between-subject (**c**) network. Edge weight indicates the inclusion proportions of each edge in 1000 times case-drop bootstrap (25%). Blue and red edges indicate positive and negative associations, respectively.

**Table 1 ijerph-18-05149-t001:** Work-related basic needs satisfaction items (2.2.1).

Construct	Mean (*SD*), Range, *α*	Sample Item
Autonomy		I feel free to do my job the way I think it could best be done.
At the office	4.00 (*0.85*), 1–5, 0.88	
Working from home	4.04 (*0.78*), 1–5, 0.86	
Competence		I feel competent at my job.
At the office	3.85 (*0.70*), 1–5, 0.81	
Working from home	3.77 (*0.73*), 1–5, 0.86	
Relatedness		At work, I feel part of a group.
At the office	4.14 (*0.79*), 1–5, 0.89	
Working from home	2.68 (*0.97*), 1–5, 0.91	

**Table 2 ijerph-18-05149-t002:** Resources of the individual and the job (Section 2.2.2).

Construct	Mean (*SD*), Range, *α*	Sample Item
Role clarity	4.57 (*1.00*), 1–7, 0.88	In my job, I always now exactly in what way to accomplish my tasks.
Job control	3.33 (*0.54*), 1–4, 0.83	Can you decide for yourself when to finish a job?
Social support	3.28 (*0.57*), 1–4, 0.84	How easy is it to talk to your colleagues?
Emotional exhaustion	2.61 (*0.92*), 1–6, 0.86	I feel frustrated by my job
Segmentation preference	4.86 (*1.68*), 1–7, 0.94	I care to separate work and private life.
Reappraisal	4.62 (*1.11*), 1–6, 0.88	When I want to feel more positively, I try to appraise the situation differently.

**Table 3 ijerph-18-05149-t003:** Daily measures of well-being and motivation (Section 2.2.3).

Construct	Mean (*SD*), Range, *α*	Sample Item
Positive affect	3.17 (*0.82*), 1–5, 0.91	Today, I felt excited.
Negative affect	1.54 (*0.65*), 1–5, 0.89	Today, I felt distressed.
Detachment from work	3.58 (*1.05*)*,* 1–5, 0.94	Tonight, I could forget about work.
Flow	4.15 (*1.41*)*,* 1–7, 0.90	Today at work, I did not notice the time passing.
Work engagement	4.12 (*1.32*)*,* 1–7, 0.96	Today, my job inspired me.

**Table 4 ijerph-18-05149-t004:** Linear regression models regressing background variables onto satisfaction of each need when working from home.

	Autonomy	Competence	Relatedness
Intercept	4.06 ***	3.79 ***	2.70 ***
Autonomy pre-pandemic	0.22 **	−0.21 **	−0.15
Competence pre-pandemic	0.02	0.45 ***	0.04
Relatedness pre-pandemic	−0.07	−0.11 *	0.01
Job control	0.16 **	0.14 **	0.06
Role clarity	0.03	0.04	−0.10
Social support by colleagues	0.06	0.08	0.30 ***
Segmentation preference	−0.09	−0.07	−0.13
Emotional exhaustion	0.03	0.04	0.13
Reappraisal	0.18 ***	0.14 **	0.19 *
Age	0.05	0.10 *	0.14 *
Childcare	−0.07	−0.06	−0.10
*R*^2^/*R*^2^ adjusted	0.378/0.338	0.454/0.419	0.158/0.104

* *p* < 0.05, ** *p* < 0.01, *** *p* < 0.001.

**Table 5 ijerph-18-05149-t005:** Multilevel model of change for the once-daily assessments of work engagement, detachment, and flow, as well as positive and negative affect. The time trend describes the linear trend over the 10 workdays/2 weeks the study spanned, and the levels of autonomy, competence, and relatedness are those reported for work from home. *n* = 1652 observations, clustered in 192 individuals.

	Work Engagement	Detachment	Flow	Positive Affect	Negative Affect
*Predictors*					
Intercept	3.87 *	3.47 *	3.94 *	3.07 *	1.69 *
Time trend	0.05 *	0.02 *	0.04 *	0.02 *	−0.03 *
Autonomy	0.08	−0.01	0.02	0.00	−0.03
Competence	0.40 *	0.13	0.42 *	0.23 *	−0.13 *
Relatedness	0.11	0.00	0.06	0.06	0.00
*Random effects*					
σ^2^	0.95	0.58	1.14	0.4	0.22
τ_00_	0.57	0.55	0.66	0.22	0.19
ICC	0.38	0.49	0.37	0.36	0.46
*Model fit*					
AIC	4937.19	4219.73	5245.81	3484.38	2554.35
BIC	4985.87	4268.42	5294.49	3533.07	2603.04
*R*^2^ (marginal)	0.147	0.017	0.107	0.101	0.063
*R*^2^ (conditional)	0.470	0.496	0.434	0.423	0.497

* = *p* < 0.01.

## Data Availability

Data will be made available upon publication here: https://osf.io/r2j4c/files/ (as of May 2021).

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
