# Peer review of "Having to Work from Home: Basic Needs, Well-Being, and Motivation"

_ijerph, 2021, doi:10.3390/ijerph18105149_

Round 1
Reviewer 1 Report
Dear Authors,
I appreciate the efforts that went into this paper. Thank you very much for give me the opportunity to review it also. This paper "Having to work from home: work engagement, well-being, and basic need satisfaction" which reveals important aspects related to the work development during pandemic from home.
On the other hand, the paper needed some changes I would like to give more specific recommendations for the authors' that I believe will improve the paper. I think this paper will fit well in the journal after the revisions.
TITLE
Satisfaction construct is not considered in Table 1. I propose to review the title without include the word satisfaction. The title should be coherent with the measured constructs or, controversially, you need to review the Table 1 constructs.
ABSTRACT
You do not specify where the sample comes from.
KEYWORDS
More possible keywords can be included, i.e. well-being at work, work in pandemic, among others. Please, review.
INTRODUCTION
You refer only to the lockdown in Germany, but many countries have closed schools. I suggest including other countries as examples and, finally, explaining the situation in Germany.
Although the goals have been established in the introduction, the objectives and, if there are any hypotheses, should be included as the last paragraph of the introduction.
MATERIALS AND METHODS
Sample has not been calculated. It must be clearly justified if there is representativeness in the selected sample.
I propose to include a table with the sociodemographic characteristics of the sample.
Constructs of Table 1 need to be clarified and explained. I suggest to make a paragraph to demonstrate consider studies proposed as citation.
2.2.1 and 2.2.2 should be included in Table 1 if they used.
Panel analysis is explained, but total data and periods are not mentioned. Please, justify how many periods (daily periods) have you included.
The items of the questionnaire should be presented.
Scale coefficients neither are nor mentioned.
If it would be necessary, the reliability (among other coefficients to demonstrate the realibilty) of the measurement scale could be included.
RESULTS
Table 2 includes some constructs which are not proposed in the Table 1. Age and Child care are not commented previously, among others. Please, review and include all the constructs in MATERIALS AND METHODS. Autonomy, Competence and Relatedness are not explained previously. Please, define and clarify.
I suggest to present R2 adjustment in other line. Please, present other information of the model such as Test F and constant. If you propose an estimation of panel data other criteria could be included, i.e. Akaike and Bayesian information criteria(AIC and BIC) to provide information about likehood. Please, review.
Variables and determinants of the 2.3. are not explained in MATERIALS and METHODS.
DISCUSSION
Some data commented in the discussion is not mentioned in the methods (i.e. line 493: 20 days of work…).
Childcare is included in discussion, but not commented in the previous test.
Limitations of the study are not clarified. Please, distingue in the paragraphs.
In general: Review the discussion and extract the parts that are not previously indicated in the test.
The discussion could be arranged in sub-chapters, in this way, the reading is facilitated and the ideas are reorganized according to the objectives, hypotheses and results.
Finally, future studies and strategies (i.e. political, social, etc…) are not commented.
Author Response
Reviewer 1 – Responses to te Reviewer’s points
POINT 1: I appreciate the efforts that went into this paper. Thank you very much for give me the opportunity to review it also. This paper "Having to work from home: work engagement, well-being, and basic need satisfaction" which reveals important aspects related to the work development during pandemic from home.
On the other hand, the paper needed some changes I would like to give more specific recommendations for the authors' that I believe will improve the paper. I think this paper will fit well in the journal after the revisions.
RESPONSE 1: We very much appreciate the overall positive evaluation of our work and its fit to IJERPH, as well as the constructive feedback. We did our best to incorporate the suggested changes and hope to have improved the manuscript in doing so.
POINT 2: Satisfaction construct is not considered in Table 1. I propose to review the title without include the word satisfaction. The title should be coherent with the measured constructs or, controversially, you need to review the Table 1 constructs.
RESPONSE 2: We apologize the confusion that Table 1 cause, see also our response to point #8. It was intended only as a concise summary of the background variables that we did not assume to change due to the pandemic, but rather to be resources hampering or helping their experience of autonomy, competence, and relatedness, i.e. the three components of basic needs’ satisfaction. The latter were not included in the table, as the table was supposed to include only ‘independent’ variables. We have deleted the table and instead included the information under 2.2.1. Also, we followed the suggestion and deleted ‘satisfaction’ from the title to avoid any unnecessary confusion – we also added motivation to the title, to underline the link to social determination theory. ‘Basic needs’ is the umbrella term that describes autonomy, competence, and relatedness to colleagues, and thus, ‘basic need satisfaction’ describes how well these needs were met – however, we do agree that the title does not require the word ‘satisfaction’ in it, as ‘basic needs’ makes perfectly clear what is assessed; and a shorter title is not a bad idea.
POINT 3: ABSTRACT
You do not specify where the sample comes from.
RESPONSE 3: Thanks for the catch! We added that the sample consisted of German employees (l. 10, 36, 239).
Point 4: KEYWORDS
More possible keywords can be included, i.e. well-being at work, work in pandemic, among others. Please, review.
RESPONSE 4: We thank Reviewer 1 for the nudge to include more keywords; we have added “well-being at work”, “work in the pandemic” as well as “adjustment to COVID-19 measures”.
POINT 5: INTRODUCTION
You refer only to the lockdown in Germany, but many countries have closed schools. I suggest including other countries as examples and, finally, explaining the situation in Germany.
RESPONSE 5: Thank you very much for this valuable suggestion which helps to put our study into context better. We now dedicate more sentences at the beginning of the introduction to describing the situation in which the study took place, and compare the German situation to those of the bigger European neighbours, of which most issued actual lockdowns, while Germany opted for a lighter version of measures, closing schools, theatres and restaurants, but not imposing curfew or confinement but mainly relying on self-compliance with social distancing rules. Thus, the situation was a lesser disruption of routines and restriction of personal freedom than in many other countries, which leads us to expect our results to be more comparable to ‘regular’ work-from-home situations than a conducted at the same time in, say, Italy or France. See line 26 – 65.
POINT 6: Although the goals have been established in the introduction, the objectives and, if there are any hypotheses, should be included as the last paragraph of the introduction.
RESPONSE 6: This is a very valuable suggestion that helps readers what to expect in the following. This fits nicely with Reviewer 2’s suggestion to re-order the introduction so that the reader is better prepared for what to expect in the later parts of the manuscript. We have thus added objectives and hypotheses at the end of the introduction, line 93 - 225.
POINT 7: MATERIALS AND METHODS
Sample has not been calculated. It must be clearly justified if there is representativeness in the selected sample. I propose to include a table with the sociodemographic characteristics of the sample.
RESPONSE 7: We thank Reviewer 1 for the nudge to provide more information about our sample, and to make more clear where to find this information: line 279-305. We have thus made a separate paragraph (the last one) in the participants and procedure section, and refer to this in the method section. All variables that describe the sample are flagged in italics.
POINT 8: Constructs of Table 1 need to be clarified and explained. I suggest to make a paragraph to demonstrate consider studies proposed as citation.
2.2.1 and 2.2.2 should be included in Table 1 if they used.
RESPONSE 8: We excuse the confusion Table 1 caused. It was our intention to avoid a lengthy paragraph on backround variables in the method section, but it deems us now that it cannot be clarified only by introducing Table 1 better. We have thus deleted Table 1 and included its information, including citations, in the Method Section under 2.2.2 “Background questionnaires: personal and organizational resources”, and thus adjusted the format to that of 2.2.1 (basic needs) and 2.2.3 (daily measures). We are convinced that this makes it more clear that there were three types of variables assessed: background variables (2.2.2) that were assessed once, work-related basic needs (2.2.1) twice – for office times and the work from home situation, as well as the daily measures (2.2.3) which were assessed daily, i.e. up to 10 times.
POINT 9: Panel analysis is explained, but total data and periods are not mentioned. Please, justify how many periods (daily periods) have you included.
RESPONSE 9: The requested information is to be found in Table 3, “Multilevel model of change for the daily assessments of work engagement, detachment and flow as well as positive and negative affect. The time trend describes the linear trend over the 10 work days / 2 weeks the study spanned, and the levels of autonomy, competence, and relatedness are those reported for work from home. N = 1652 Observations, clustered in 192 individuals.” We have followed the reviewer’s suggestion and included the information that answers were given once daily (L. 278 and L. 549).
POINT 10: The items of the questionnaire should be presented. Scale coefficients neither are nor mentioned. If it would be necessary, the reliability (among other coefficients to demonstrate the realibilty) of the measurement scale could be included.
Table 2 includes some constructs which are not proposed in the Table 1. Age and Child care are not commented previously, among others. Please, review and include all the constructs in MATERIALS AND METHODS. Autonomy, Competence and Relatedness are not explained previously. Please, define and clarify.
RESPONSE 10: We appreciate very much that the reviewer 1 pointed out that we never mentioned sample items for our main variables of the work-related needs, and were thus very happy for the suggestion and immediately included them in the method section on the work-related basic needs (2.2.1). Age and Childcare are part of the demographics, thus can be found in the sample description, see 2.1 PARTICIPANTS AND PROCEDURE, line 279-305 .
Point 11: I suggest to present R2 adjustment in other line. Please, present other information of the model such as Test F and constant. If you propose an estimation of panel data other criteria could be included, i.e. Akaike and Bayesian information criteria(AIC and BIC) to provide information about likehood. Please, review.
Response 11: We thank the reviewer for this nudge and following the suggestion, have included R2 in another line section of the table called model fit, which features BIC and AIC at its top (see Table 2).
Point 12: Variables and determinants of the 2.3. are not explained in MATERIALS and METHODS. Some data commented in the discussion is not mentioned in the methods (i.e. line 493: 20 days of work…). Childcare is included in discussion, but not commented in the previous test.
Response 12: We kindly refer to our PARTICIPANTS AND PROCEDURE section (2.1, line line 279-305), where we present all data describing the demographics of our sample, including its average age, childcare duties, and the (approximately 20) days spent working from home before entering the study.
Point 13: Limitations of the study are not clarified. Please, distingue in the paragraphs.
RESPONSE 13: We are sorry that we did not flag the mentioned limitations appropriately. We have now introduced paragraphs with subheadings in the discussion, and labeled one of them ‘Limitations’, where we describe the main shortcomings of the study, followed by ‘however’, introducing our arguments why the mentioned shortcomings only limit, but do not negate, the merit of the study or proposed interpretation of its results.
POINT 14: In general: Review the discussion and extract the parts that are not previously indicated in the test.
The discussion could be arranged in sub-chapters, in this way, the reading is facilitated and the ideas are reorganized according to the objectives, hypotheses and results. Finally, future studies and strategies (i.e. political, social, etc…) are not commented.
RESPONSE 14: We thank the reviewer for this great idea to improve the readibility of the admittedly lenghty discussion – we have arranged it in terms of subsections, which include not only limitations, but also future studies and interventions.

Reviewer 2 Report
This paper described a study on employed people during the initial period of the COVID-19 pandemic (April and May 2020 in Germany) to examine adaptation to working from home from working at work. Overall, I thought the results were interesting given the modeling techniques used. My main concerns are in the opening of the paper. Specifically, the introduction does not currently have a logical flow. That is, you briefly introduce us to your topic, state your goals (paragraph 2), and then start defining terms and parts of SDT. Then you list another set of expectations and research questions (paragraph 4). This needs to be revised/reorganized for a better read as I was thoroughly confused until I got to the methods and results to see what was actually done and found.
Author Response
REVIEWER 2 – RESPONSE TO THE REVIEWER’S COMMENTS
POINT 1: This paper described a study on employed people during the initial period of the COVID-19 pandemic (April and May 2020 in Germany) to examine adaptation to working from home from working at work. Overall, I thought the results were interesting given the modeling techniques used.
RESPONSE 1: Thanks!
POINT 2: My main concerns are in the opening of the paper. Specifically, the introduction does not currently have a logical flow. That is, you briefly introduce us to your topic, state your goals (paragraph 2), and then start defining terms and parts of SDT. Then you list another set of expectations and research questions (paragraph 4). This needs to be revised/reorganized for a better read as I was thoroughly confused until I got to the methods and results to see what was actually done and found.
RESPONSE 2: We thank Reviewer 2 for the nudge to improve the logical flow of our introduction. We have made an effort and hope it flows better now, first describing the situation which motivated the study (Covid-19 measures in Germany), then explaining the importance of basic needs for intrinsic motivation as per Self-Determination Theory, and then lay out in detail our research objectives and hypotheses.
Reviewer 3 Report
The study investigates a relevant and urgent issue, which deserves an accurate analysis of a phenomenon that is going to change how we design, manage and live our work life. The research question is well posed, it is clear, well-grounded with literature references. The research method is clear as well, and the data analysis is accurate. The discussion takes into account the results and provides a reasonable explanation of the findings.
There are some aspects I’d like to share with the authors, which may deserve some further explanation. Here is the list of my comments:
L 9: “the well employees” seems like a typo
L 100: the data collection started in April, how long after the beginning of lockdown? This may be a relevant aspect, since workers may already have had the time to adjust to the new work condition
L 112: the authors used a retrospective question for evaluating the basic needs satisfaction, however they never mention potential biases for such method
L 117: two weeks seems like a short time span for some needs to settle down. How did the authors compute this time window? On what scientific evidence?
L 140: children’s age and autonomy may have a great impact on work management, therefore it should be taken into account. Did the authors collect children’s age and autonomy?
L 144: The fact that 76% of the sample reported 0 childcare duties is a particularly relevant aspect, which should be taken into account in the discussion of results, because childcare is recognized to be one of the most impactful factors on work-from-home tasks
Table 1: there is a typo: “In my job, I always know…”
L 162: it is not clear how the authors investigated the basic needs. Did they use a specific questionnaire?
L 190: this sentence is not clear, especially the last one (line 192)
L 221: typo: “psychometric”
L 396: It may also depend on the kind of work tasks and the previous experience on technology mediated work. For instance, many teachers and professors may have experienced a critical loss of competence, passing abruptly from lessons in person to lessons online, since they did not have solid expertise with the technological tools and the method of online teaching.
Did the authors take into account the kind of tasks performed by the sample? Was it administrative? or managerial? or other forms of tasks?
L 403: I agree, therefore the authors should have taken into account the organizational and managerial context, which may have facilitated the abrupt transition to work from home. Any data on that?
L 418: I think this aspect should be explained clarifying the two intertwined phenomena: the condition of working from home and the sudden, forced transition from one modality to the other. The sample under investigation was experiencing both the abrupt transition and the new work condition, therefore no clear inference could be drawn about working from home in general. The autonomy experience may be reduced because the company had to manage the abrupt transition consistently across all the workers.
L 425: this sentence is not very clear, could be rewritten
L 432: “It is reassuring that working from home, with most often implies…” perhaps the authors meant “which”?
Author Response
REVIEWER 3 – RESPONSES TO REVIEWER’S COMMENTS
POINT 1: The study investigates a relevant and urgent issue, which deserves an accurate analysis of a phenomenon that is going to change how we design, manage and live our work life. The research question is well posed, it is clear, well-grounded with literature references. The research method is clear as well, and the data analysis is accurate. The discussion takes into account the results and provides a reasonable explanation of the findings. There are some aspects I’d like to share with the authors, which may deserve some further explanation.
RESPONSE 1: We thank Reviewer 3 for the praise and hope to were able to further improve the paper by incorporating the reviewer’s helpful suggestions.
POINT 2: L 9: “the well employees” seems like a typo
RESPONSE 2: Thanks for the catch! We adjusted it to “how well employees worked and felt”.
POINT 3: L 100: the data collection started in April, how long after the beginning of lockdown? This may be a relevant aspect, since workers may already have had the time to adjust to the new work condition
RESPONSE 3: Indeed this is a relevant aspect. It is considered in our variable “days working from home” when entering the study, which is on average 20 days (line 297) and discussed in line 772 ff. . As in Germany, there was no official lockdown – we have removed any wording suggesting otherwise – there was no universal starting point from which any employee started working from home, some started earlier than others. However, this variable did not influence the level or slope of our daily indicators (l. 776).
POINT 4 - L 112: the authors used a retrospective question for evaluating the basic needs satisfaction, however they never mention potential biases for such method
RESPONSE 4: We agree with the reviewer that asking for a retrospective rating is never the gold standard of information, but sometimes the only way to get any data at all on a point in time before an unexpected change in the situation, which is why we recurred to this. We followed the reviewer’s suggestion in adding that this estimation might be biased by recall bias, adding references no. 8 and 9 on recall bias, see line 256ff.. However, we see no reason to assume systematic bias in a certain direction across the board in our whole sample so that average estimates would be systematically skewed). In addition, we refer to data we gathered before the pandemic, where other samples reported on the same constructs – average means were comparable to those of the retrospective reports of office-time need satisfaction (see 2.2.1).
POINT 5: L 117: two weeks seems like a short time span for some needs to settle down. How did the authors compute this time window? On what scientific evidence?
RESPONSE 5: We agree that two weeks are short for settling into a new situation, however, it is long for partaking in a study – in order to prevent massive dropout, we reduced the study’s duration to two weeks, a typical convention in daily diary research (while many studies are even limited to only one work week, i.e. 5 days), see line 274f.. Bear in mind that we did not capture anyone’s development from the first day working from home, but everyone in the sample had already settled in to a certain extent – that means, what we were witnessing was a snippet of a long total adjustment phase that had begun earlier and that spanned much longer, even into the second and third lockdown in many countries. Any positive trend over the course of the two weeks we were witnessing could be counted as indicator of an overall positive adjustment.
POINT 6: L 140: children’s age and autonomy may have a great impact on work management, therefore it should be taken into account. Did the authors collect children’s age and autonomy?
RESPONSE 6: No, sorry. Childcare was not at all at the focus of the present paper, but a mere control variable. We do acknowledge the importance of childcare though by referring to studies that were investigating the topic more thoroughly (26-28, might change).
POINT 7: L 144: The fact that 76% of the sample reported 0 childcare duties is a particularly relevant aspect, which should be taken into account in the discussion of results, because childcare is recognized to be one of the most impactful factors on work-from-home tasks
RESPONSE 7: Yes, please find its discussion in line 710 ff., where we also refer to studies that actually put their focus on childcare and home-to-work interference in the pandemic (26-28).
POINT 8: Table 1: there is a typo: “In my job, I always know…”
RESPONSE 8: Thanks for the catch.
POINT 9: L 162: it is not clear how the authors investigated the basic needs. Did they use a specific questionnaire?
RESPONSE 9: Thanks for the catch; we have included the citation and sample items in 2.2.1, l. 323 ff.
POINT 10 - L 190: this sentence is not clear, especially the last one (line 192)
RESPONSE 10: Thanks for nudging us to be clearer; the sentence now reads “Need satisfaction when a) working from home (at the moment) vs. b) when still working in the office (before the pandemic) was compared with paired t-tests for dependent samples.”
POINT 11 - L 221: typo: “psychometric”
RESPONSE 11: We agree that this confused us first, too – the package is called “psychonetric”, because it is a psychometric approach to model NETworks – a wordplay on part of the package’s author Sascha Epskamp.
POINT 12: L 396: It may also depend on the kind of work tasks and the previous experience on technology mediated work. For instance, many teachers and professors may have experienced a critical loss of competence, passing abruptly from lessons in person to lessons online, since they did not have solid expertise with the technological tools and the method of online teaching.
Did the authors take into account the kind of tasks performed by the sample? Was it administrative? or managerial? or other forms of tasks?
L 403: I agree, therefore the authors should have taken into account the organizational and managerial context, which may have facilitated the abrupt transition to work from home. Any data on that?
RESPONSE 12: While we have no information on the managerial context that may have facilitated the transition, we have very broad information on the type of tasks and fields people worked on / in. No one of our sample reported teaching (nor nursing, or arts) to be their primary kind of task, most people reported working at the computer or with clients as their primary to-do. In terms of industry, most people reported working for the public sector, IT/communication, and science. As these categories are rather broad and of little meaning, we have abstained from interpreting it – we apologize for not picking better categories when setting up the study. We included this information very shortly in the description of the sample, i.e., Participants and Procedure, line 296f.
POINT 13: L 418: I think this aspect should be explained clarifying the two intertwined phenomena: the condition of working from home and the sudden, forced transition from one modality to the other. The sample under investigation was experiencing both the abrupt transition and the new work condition, therefore no clear inference could be drawn about working from home in general. The autonomy experience may be reduced because the company had to manage the abrupt transition consistently across all the workers.
RESPONSE 13: We absolutely agree with the Reviewer, therefore we have elaborated (l. 430 ff.) on disentangling the different dynamics of April and May of 2020, work from home, an abrupt transition, and a pandemic that Please compare with our discussion of this point in the last paragraph of the discussion. (l.811ff., particularly 816f.).
POINT 14: L 425: this sentence is not very clear, could be rewritten
RESPONSE 14: We have adjusted the sentence.
POINT 15 - L 432: “It is reassuring that working from home, with most often implies…” perhaps the authors meant “which”?
RESPONSE 15: Thanks for the catch – we corrected that.

Round 2
Reviewer 1 Report
Please, I propose to present the 2.2. Measures (2.2.1, 2.2.2 and 2.2.3) analysis in tables.
Author Response
Dear Reviewer,
thank you very much for your suggestion. Please find attached a revised version of the manuscript, including the tables for 2.2.1, 2.2.2, and 2.2.3. We hope that in this version, the manuscript will find your approval.
Kind regards and best wishes,
Hannah Schade